# The Role of Dandelion (*Taraxacum officinale*) in Liver Health and Hepatoprotective Properties

**DOI:** 10.3390/ph18070990

**Published:** 2025-07-01

**Authors:** Francisca Herrera Vielma, Matías Quiñones San Martin, Nicolás Muñoz-Carrasco, Fernanda Berrocal-Navarrete, Daniel R. González, Jessica Zúñiga-Hernández

**Affiliations:** 1Pharmacology Laboratory, Biomedical Sciences Department, Faculty of Health Sciences, University of Talca, Talca 3465548, Chile; francisca.ingbiotec@gmail.com (F.H.V.); mquinones@utalca.cl (M.Q.S.M.); nicolas.munoz.01@alu.ucm.cl (N.M.-C.); fberrocal18@alumnos.utalca.cl (F.B.-N.); dagonzalez@utalca.cl (D.R.G.); 2Doctorate Program in Science, Mention in Research and Development of Bioactive Products, Institute of Chemistry of Natural Resources, University of Talca, Talca 3465548, Chile; 3School of Biotechnology Engineering, Faculty of Agricultural and Forestry Sciences, Catholic University of Maule, Talca 3466706, Chile; 4School of Biochemistry, Institute of Cell Biology and Biotechnology, University of Talca, Talca 3466706, Chile

**Keywords:** bioactive compounds, traditional medicine, taraxasterol, antioxidant, liver fibrosis, chronic liver diseases

## Abstract

**Background:** *Taraxacum officinale* (*T. officinale*), commonly known as dandelion, is a plant with recognized therapeutic properties in both traditional and modern medicine. Historically, it has been used to treat various conditions, particularly liver disorders, owing to its antioxidant and anti-inflammatory activities. This narrative review focuses on its biological activity, with an emphasis on hepatoprotective effects. **Methods:** We performed a compilation and analysis of published studies on the effects of *T. officinale* in animal models and its potential application in liver diseases. **Results:** Preclinical studies have reported that extracts of this plant protect against liver damage induced by toxic agents such as alcohol, carbon tetrachloride, and paracetamol. Among the most relevant and predominant bioactive compounds of *T. officinale* is taraxasterol, which modulates inflammatory and oxidative stress pathways, helping to prevent liver damage. **Conclusions:** While preclinical studies are promising, further clinical trials are essential to confirm the safety and efficacy of *T. officinale* in the treatment of liver diseases. Determining the optimal dosing, evaluating its potential as an adjuvant in pharmacological treatments, as well as evaluating possible interactions with conventional drugs, is necessary for the potential use of *T. officinale* as an adjuvant agent in the treatment of liver diseases.

## 1. Introduction

Medicinal plants have been used since ancient times. Different civilizations have gained knowledge about the medicinal properties of a large number of plants found in their environment. For centuries, plant species used as sources of food and medicine have shown therapeutic properties, produced by substances or active principles found in their constituents, with protective effects against diseases [1]. Historically, communities worldwide have depended on herbal preparations to treat various health disorders. Nowadays, the World Health Organization (WHO) has promoted the use of traditional medicinal plants for the management and prevention of several diseases [2]. In 1997, the WHO indicated that a medicinal plant is “*any plant in which one or more of its organs contain substances that can be used for therapeutic purposes or as precursors in the synthesis of other drugs*”. The WHO description discriminates between plants/herbs whose compounds and beneficial properties have been scientifically well studied, from those that, despite being considered medicinal, have not yet been fully assessed. Among them, *Taraxacum officinale* Weber ex F.H.Wigg [3] has recently received scientific attention due to its high nutritional value and its beneficial effects on human health [4]. *Taraxacum officinale* (*T. officinale*) is commonly known as “dandelion”, a plant of the genus *Taraxacum* from the *Asteraceae* family. *T. officinale* is a non-toxic perennial herbaceous plant frequently considered a weed [5] that can be exploited due to its beneficial properties [6]. The common name “dandelion” probably comes from the French “*dent-de-lion*”, which refers to the “*toothed*” leaves of the plant [7]. As Stearn defined in 1973, the coin “officinale” literally means “be appropriate to an office”, indicating the storehouse of a monastery, where “medicines and other necessary items were lay-up”. Therefore, this spice designation is coined for plants that have recognized medicinal, gastronomic, or other uses [8].

## 2. Botanical, Ethnopharmacological, and Experimental Overview of *Taraxacum officinale*

### 2.1. Distribution, Habitat, and Botanical Description

It is thought that *T. officinale* emerged in Eurasia, from where it was involuntarily disseminated by population movement, spreading it almost all over the world [9]. *T. officinale* can tolerate a varied range of environmental conditions, growing in optimal conditions in well-drained soils, enriched in calcium and high humus content [10]. It grows in grasslands, roadsides, parks, orchards, and vacant lots, adapting to light and growing strongly in spaces with direct sunlight, although it can also grow in indirect or diffuse light. Additionally, it tolerates drought seasons and frost [11].

*T. officinale* is a perennial herbaceous plant, about 10–50 cm high. It has a tapering, brown, and poorly branched root, which releases a white and bitter latex (Figure 1A). The plant has a radical rosette of simple, lobed leaves with irregular teeth, arranged in a rosette, light to dark green in color and bitter in flavor, about 40 cm tall, and 0.7 to 15 cm wide (Figure 1B). The basal rosette produces hollow, cylindrical stems, topped with large, solitary yellow flowers (on average, 5 to 10 flowers) that close at dusk and when rain is approaching (Figure 1C). The fruits/seeds are tapered achenes of grayish-brown color and a white umbrella-shaped papilla (Figure 1D) [12].

### 2.2. History and Traditional Uses of T. officinale

*T. officinale* has been used since ancient times for the treatment of several health disorders. Theophrastus (371 BC-287 BC), a Greek naturalist, was one of the first to describe this plant. He recommended it to be used as a stimulant, especially against freckles and other liver-related skin conditions [13]. In the past, it was often consumed fermented into wine. Celts and Anglo-Saxon tribes used it to prevent scurvy, as a diuretic, and a laxative. Traditional Chinese medicine applied *T. officinale* roots for the treatment of swelling. Also, *T. officinale* juice was recommended for the protection of the liver and against dropsy (fluid retention in one or more areas of the body), as well as an antidote against scorpion stings [14]. In the 10th and 11th centuries, Arab physicians used it to treat ailments of the liver and spleen [15]. In 1543, Leonhard Fuchs, a botanist, defined its medicinal use in gout, diarrhea, blisters, liver conditions, among others. In native North American medicine, they used *T. officinale* root-based concoctions and cooking to treat kidney disease, stomachache pains, and heartburn [16]. It has also been used in folk medicine as a “blood purifier and laxative”, for skin conditions such as eczema, and for the treatment of arthritic and rheumatic conditions [7]. In Mexico, a whole herb decoction is traditionally used to control diabetes mellitus. In the case of traditional Chinese medicine, its main use is in combination with other herbs to treat hepatitis, improve the immune response to infections of the upper respiratory tract, such as bronchitis or pneumonia, and at a folk level, it is also used in compresses due to its anti-mastopathic activity. All of the above indicate that Chinese culture considers this a non-toxic plant [17]. In the 19th century, studies began to find scientific explanations related to the mechanism of action of *T. officinale* relative to its multiple beneficial actions: antioxidant, cholagogue, anti-inflammatory, analgesic, anticoagulant, choleretic, angiogenic, and anticancer, among others [18]. Currently, all its parts (roots, foliage, and flowers) are commercially available in different pharmacological and supplemental preparations that are suggested to treat, for example, some liver, gallbladder, and kidney disorders. Since the 1950s, biochemical composition and bioactive compounds identification of *T. officinale* have been described. A common use is as an antioxidant, due to its free radical scavenging activity, as well as the anticancer effect of the root [19]. This might lead to *T. officinale* and its bioactive derivatives being established as interesting therapeutic tools.

Therefore, the aim of this narrative review is to critically synthesize the current preclinical evidence regarding the hepatoprotective effects of *T. officinale*, focusing on its bioactive compounds, mechanisms of action, and potential applications in liver disease. Unlike previous reviews that broadly cover the general medicinal uses of dandelion, this work specifically highlights its role in liver protection and metabolic liver disorders, emphasizing recent findings published in the last two decades.

### 2.3. Chemical Composition of T. officinale

The characterization of *T. officinale* composition has been a subject of interest, mainly to find bioactive components associated with its pharmacological properties [20]. *T. officinale* is rich in vitamins, inulin, and is a rich source of phytochemicals, including terpenes, phenolic acids, and flavonoids, alongside being a good source of amino acids and minerals, particularly potassium [14]. In addition, its bitter substances are known to stimulate gut function, while phenolic derivatives have anti-inflammatory and antioxidant properties, making the nutritional composition of *T. officinale* of high interest when used as food and fodder [21,22]. Research on *T. officinale* bioactive compounds, particularly phenols, terpenes, and flavonoids, accounts for almost half of all research supporting the importance of this plant in providing health and medical benefits, reducing the high cost, and rethinking the technological/chemical approaches to disease treatments currently used in medical care [18]. For this reason, it is necessary to know the organic compounds and some examples of their respective biological activity, which are presented in Table 1.

### 2.4. In Vitro and In Vivo Evidence of T. officinale Biological Activities

To date, most research on *T. officinale* has reported the biological activity of plant extracts. A literature search on the use of *T. officinale* in various models of diseases and disorders is summarized in Table 2, which highlights the versatility of the different plant extracts and fractions. These fractions confer metabolic benefits (diabetes, obesity, dyslipidemia), protect against bone pathologies and bacterial infections associated with counteracting oxidative processes, and inhibit tumor growth in experimental models. Notably, consistent antioxidant and anti-inflammatory properties emerge from most of these studies, suggesting that many of the reported effects are mediated through modulation of oxidative stress and inflammatory pathways. Table 3 summarizes the evidence attributing hepatoprotective properties to *T. officinale* in various pathological contexts. Over the past two decades, this species has gained prominence in biomedical research for its antioxidant, anti-inflammatory, and lipid metabolism-regulating properties. In experimental hepatology, it is currently being assessed both as a prophylactic agent promoting liver protection and regeneration and as a therapeutic agent, intervening in established lesions and even in neoplastic processes.

It is important to note that outcomes vary depending on the part of the plant used, extraction method, and experimental model; consequently, clinical extrapolation must be approached with caution. Nevertheless, the body of work compiled in these tables supports *T. officinale* as an important source of bioactive compounds with broad therapeutic applications and underscores the need for further preclinical and clinical studies, aimed at standardizing extracts, elucidating molecular mechanisms, and defining safe and effective dosing regimens.

### 2.5. T. officinale and Liver Pathologies

Chronic liver disorders (CLDs) are among the main health complications worldwide, becoming an important public health problem due to the high morbidity rate [80]. CLDs are related to malnourishment, metabolic syndrome, viral infections, alcohol, and drug abuse, among others [81]. Pharmacological treatments for the disorders mentioned above are often difficult to achieve and may have limited efficacy [82]. Consequently, complementary and alternative medicines for the treatment of liver disorders have become of special interest in recent years, as potential natural agents to diminish or avoid the risk of harm when these compounds are used clinically [83].

As mentioned above, *T. officinale* has been used for many years as the main component of various herbal preparations for hepatoprotection. The monographs of the European Scientific Cooperative on Phytotherapy (ESCOP) certify the action of the root as a restorer of liver and biliary function, and its indication for dyspepsia and loss of appetite was scientifically proven (Taraxaci Radix (Dandelion Root)) [84]. The Expert Commission of the German Ministry of Health (German Commission E) approved the traditional use of leaves and roots with aerial parts to activate urinary elimination, as an adjuvant in mild urinary complaints, relief of digestive disorders, as well as for loss of appetite [9]. The European Medicines Agency (EMA) supports the traditional use of *T. officinale*, particularly leaves and root extracts, to improve the elimination of body fluids. The EMA also approved the traditional consumption of aerial parts for the relief of mild gastrointestinal disorders, such as satiety, flatulence, and slow digestion. It is also indicated as a stimulant of liver function [59]. Scarce data are available on preclinical and clinical studies with this plant extract. As the concept of reverse pharmacology is rapidly growing from the discovery of ‘leads’ from plants, it is worthwhile conducting scientific studies on *T. officinale* to support its traditional use.

### 2.6. In Vivo Liver Studies of T. officinale Effects on Acute and Chronic Liver Disease

The hepatoprotective role of *T. officinale* has been investigated, particularly as an alternative for the treatment of hepatic diseases, gaining special momentum in the last decade. In the context of acute liver disease, Colle et al. assessed the protective effect of *T. officinale* leaf extracts (0.1 and 0.5 mg/mL) on acetaminophen (APAP)-induced hepatotoxicity [40]. These authors showed that the foliar extract reduced the levels of reactive oxygen species (ROS), decreased serum transaminases, and caused hepatic histopathological alterations. These beneficial effects were attributable to the high content of phenolic compounds. Domitrović et al. [85] showed that a *T. officinale* extract was able to modulate the regenerative hepatic capabilities through inactivation of stellate cells, decreasing liver fibrosis. Interestingly, these authors found that the extract restored the liver antioxidant capacity by normalizing the levels of Cu/Zn SOD. A similar antioxidant effect was observed in a model of sodium dichromate-induced hepatotoxicity and genotoxicity, whereas *T. officinale* reduced the DNA fragmentation in hepatocytes and re-established the antioxidant enzymes [64].

Regarding liver fibrosis, several studies evaluated the hepatoprotective effect of *T. officinale* using the carbon tetrachloride (CCL_4_)-induced liver injury model. As mentioned previously, Domitrovic’s group [85] and Huseini et al. found that hepatic microvesicular steatosis and liver damage were ameliorated at *T. officinale* doses higher than the average for humans (over 750 mg/kg/day) [86]. These results were corroborated by Al-Malki et al. [87] and Favari et al. [69] using aqueous leaf extracts and the aerial part of the plant, respectively. Also, ethanol and n-hexane extracts of *T. officinale* leaves were evaluated, showing that not only the aqueous extract could be hepatoprotective [88]. This work showed that a methanol extract significantly restored the levels of critical liver enzymes. In addition, bilirubin, lipid profile, and antioxidant enzymes were improved, thus protecting against oxidative stress. More recently, Hamza et al. [89] found that *T. officinale* root preparation (500 mg/kg, oral administration) presented anti-inflammatory activity through the inactivation of NF-κB pathways, negatively modulating the levels of IL-1β, TNF-α, transforming growth factor-β1 (TGF-β1), and COX-2.

One relevant consideration in CLDs is cirrhosis decompensation, generating acute-on-chronic liver failure (ACLF), which is associated with more than a 20% mortality rate one month after occurrence [90]. Similar results are observed in animal models [24]. 

### 2.7. Impact on Metabolic-Related Liver Diseases

Among liver diseases, metabolic dysfunction-associated fatty liver disease (MAFLD) has gained primary importance in recent years worldwide [91]. MAFLD is often linked to overweight/obesity and associated comorbidities such as insulin resistance and diabetes [92]. It can evolve from simple steatosis to steatohepatitis, or to major complications such as fibrosis, cirrhosis, and hepatocellular carcinoma (HCC) [93]. At present, there are no pharmacological treatments approved for the prevention or management of MAFLD [94], although there are several clinical trials ongoing [95]. Given the complexity of the underlying mechanisms, monotherapies are unlikely to provide effective solutions [96], and there is an evident need for new therapeutic alternatives [94]. Several natural compounds have been shown to exert beneficial effects against metabolic liver disease [83], and *T. officinale* may become one of them [97].

A leaf extract from *T. officinale* was evaluated in a liver steatosis model [67]. This report indicated that increased insulin and fasting glucose levels were normalized, followed by a dramatic decrease in hepatic lipid accumulation, as well as body and liver weight, with a suppression of triglycerides (TG) and total cholesterol (TC), among others. The hepatoprotective results observed for *T. officinale* provide evidence for the use of leaf extract as a therapy in the treatment of fatty liver and obesity-related disorders.

Since diabetes is a metabolic disease associated with liver function [95], it is interesting to evaluate the action of *T. officinale* in this pathology. *T. officinale* leaf extract activated the antioxidant response and decreased ROS, lipid peroxidation, and nitrite levels in streptozotocin (STZ)-induced diabetic rats. The authors concluded that a *T. officinale* leaf extract could improve the antioxidant status of the liver, suggesting its usefulness in improving diabetes-induced liver injury [68]. Another *T. officinale* root aqueous extract (400 mg/kg) was reported to improve glycemic control in diabetic mice [27]. The polysaccharides present in the aqueous extract might have been responsible for these effects due to their ability to inhibit α-glucosidase and α-amylase, which would indicate that this herb can be developed as functional foods or drugs (bioactive compounds) for the prevention and treatment of diabetes due to their effective hypoglycemic effects, promising natural alternatives for the treatment of diabetes [33,98].

### 2.8. T. officinale Bioactive Compounds on Liver Pathologies

*T. officinale* is rich in saccharides, peptides, flavonoids, terpenes, and several other phytochemicals [97]. Its polysaccharides are important functional substances [99,100], with immune regulatory and antioxidant activities. Two of these polysaccharides (TOP1 and 2) obtained from *T. officinale* exhibited hepatoprotective effects in a model of liver fibrosis. Specifically, these compounds decreased AST (aspartate aminotransferase) and ALT (alanine aminotransferase), reversed reduced glutathione (GSH) depletion and nuclear factor kappa B (NF-κB) activity, and decreased expression of its regulatory inflammatory mediators iNOS, COX-2, TNF-α, IL-1β. These data suggest that TOPs may be a potential preventive or therapeutic compound against CCL4-induced liver damage [77].

Another relevant active component isolated from *T. officinale* is taraxasterol (TA), a pentacyclic-triterpenoid [101]. TA exhibits pharmacological properties: antioxidant, anti-inflammatory, and anti-tumor effects, contributing to its protective potential against a diversity of maladies. In neurons, it has been described that the antioxidant effect of TA is achieved through the nuclear factor erythroid-related factor 2 (Nrf2) signaling pathway [101]. The protective role of TA on acute liver disease has also been studied in APAP models, finding that TA reduces liver damage associated with a decrease in ALT, AST, LDH, and ROS, normalizing the antioxidant response (catalase (CAT) and GSH). These hepatoprotective effects were mediated by activation of the Nrf2/HO-1 pathway [61]. Comparable results were also observed when acute liver injury was generated by concavalin A (Con A). TA reduced the hepatic damage generated by Con A through anti-inflammatory pathways, in particular a reduction in the activity of toll-like receptors 2 and 4 (TLR2 and TLR4) and NF-κB. Interestingly, TA prevented apoptosis by decreasing the pro-apoptotic protein Bcl-2-associated X protein (Bax) and increasing the anti-apoptotic protein leukemia protein 2 (Bcl-2), which reduced apoptosis in liver cells, demonstrating to be a potent hepatoprotective molecule [102]. TA also exerts a hepatoprotective role on ethanol-induced liver disease, mainly mediated by upregulating antioxidant defense mediated by i) antioxidant improvement due to activation of Nrf2: ROS reduction with decreasing of nitric oxide (NO), and increasing CAT, superoxide dismutase (SOD) and GSH activity; ii) anti-inflammatory actions: inhibition of the degradation of α inhibitor kappa B (Iκ-Bα), and the expression level of NF-κB, along with the reduction in the inflammatory pathway TLR4/MyD88/NF-κB, both of them associated with a depletion of pro-inflammatory TNF-α, IL-6, and IL-1β cytokines [31,103]. These findings indicate that TA may offer protective effects against ethanol-induced liver injury in mice by regulating inflammation and oxidative stress. To clarify the mechanism of action of TA, He et al. performed an RNA sequencing analysis of liver tissues subjected to fibrosis and treated with TA. Their analysis revealed that TA treatment was associated with the expression of 2675 genes related to the extracellular matrix (ECM). Specifically, TA blocked the gene expression of hypoxia-inducible factor 1 subunit alpha (HIF-1α), Smad-dependent transforming growth factor-beta (TGF)-signaling pathway (TGF-β/SMAD), and Wnt-signaling pathways, thus inhibiting the activation pathways of Ito cells, decreasing excessive ECM production [104].

Finally, the impact of TA on liver damage associated with mycotoxin exposition (aflatoxin B1) has also been investigated. It has been reported anti-inflammatory and antioxidant effects of TA are associated with inhibition of hepatocyte apoptosis by regulation of caspase-3, Bax, and Bcl-2 [63]. These authors found that TA enhanced hepatocyte autophagy by modulating the PI3K/AKT/mTOR pathway. This suggests that the mechanism of action of TA is complex and related to the activation of survival pathways. Therefore, TA could be one of the most relevant hepatoprotective agents present in *T. officinale*, but future studies regarding their safety and use as adjuvant therapies are necessary.

### 2.9. T. officinale in Liver Cancer

One of the most severe complications of liver fibrosis is hepatic cancer. Over the past decade, the incidence of liver cancer has increased by 25%, with HCC being the most prevalent form [105]. Therapeutic strategies for the treatment of HCC have experienced important advances, including surgical resection, liver transplantation, and other surgical interventions [106]. On the other hand, drug therapies against HCC, including immunotherapies, have a high rate of recurrence and resistance [107]. Alternative therapies, including herb extracts, could be of interest in this case. For this purpose, investigations have tried to elucidate the effect of TA on liver cancer. Bao et al. [62] developed an in vivo model of HCC, generated by subcutaneous injections of SK-Hep1 (liver adenocarcinoma) cells in mice and then treated orally with TA. The TA treatment inhibited tumor growth by up-regulating Histidine Triad Nucleotide-Binding Protein 1 (Hint1) and Bax expression, decreasing CyclinD1 and Bcl-2 protein expression in tumors. In the same study, TA induced cell cycle arrest in the G0/G1 phase in Hep2 cells and promoted apoptosis. Another interesting study was developed by Ren et al. [78], where they established a hepatocarcinoma model in tumor-bearing mice (mouse HCC cell line H22). In this model, the TA treatment inhibited tumor cell proliferation by depletion of Ki-67, and enhanced antitumor immunity, associated with a significant increase in CD4+ T cells in the spleen, but without effect on CD8+ T cells. As has been extensively described, hepatocarcinoma is considered an immunogenic tumor; therefore, the anti-inflammatory and antioxidant actions of *T. officinale* could be tested in the future to be used as an alternative therapeutic strategy.

Despite promising results in vitro and in preclinical studies, these hepatoprotective properties of *T. officinale* require further validation in clinical studies in humans. Both cellular and animal models help us to understand the mechanisms of action of bioactive compounds. For example, the anti-inflammatory and antioxidant properties of TA or polysaccharides are present in the plant. However, there are major differences between animal and human metabolism, which poses a challenge to clinical studies. Nowadays, there are no established clinical protocols that support the consumption of *T. officinale* as a management for liver diseases in humans, highlighting the need for rigorous clinical trials that assess safety, efficacy, and optimal dosage in human populations. This will help us to determine whether *T. officinale* can be a possible complementary or adjuvant treatment for conventional therapies, especially in chronic or acute liver pathologies where current treatments are limited.

### 2.10. Safety Considerations, Contraindications, and Interactions of Taraxacum officinale

Although *T. officinale* possesses multiple therapeutic properties, its use is not without risks. Cases of hypersensitivity have been reported in individuals allergic to plants of the Asteraceae family, which may trigger mild to moderate cutaneous or respiratory allergic reactions [108].

Due to its well-known diuretic activity, extracts of *T. officinale* may interact with antihypertensive drugs, other diuretics, and lithium salts, potentially increasing the risk of dehydration or electrolyte imbalances [109].

Furthermore, some of its bioactive compounds exhibit antiplatelet activity; therefore, its use should be approached with caution in patients taking anticoagulant medications, as it could increase the risk of bleeding [75,110].

Its use is not recommended without medical supervision in individuals with severe hepatic or renal conditions, nor in pregnant or breastfeeding women, due to the lack of conclusive clinical studies supporting its safety in these populations [111,112].

Various in vitro studies have demonstrated that *T. officinale* extracts exhibit low toxicity in normal cells, supporting its traditional use and suggesting a favorable safety profile when used at conventional concentrations. According to Schütz et al., no significant adverse effects have been reported in experimental studies [7]. However, at high concentrations, some extracts have shown cytotoxic activity against tumor cell lines, which may indicate potential antitumor effects. These findings highlight both the therapeutic potential of *T. officinale* and the need for further research, particularly in animal models and clinical trials, to confirm its safety and efficacy [7].

### 2.11. Limitations and Future Perspectives

Although the beneficial effects of *T. officinale* extracts and isolated compounds have been widely documented over the past two decades, no approved drug or therapy based on this knowledge has yet been developed. This absence may be explained by several factors, including the high variability in the chemical composition of plant extracts, which is influenced by species differences, extraction methods, and cultivation conditions. Moreover, the lack of robust and large-scale clinical trials limits the validation of their efficacy and safety in humans [7]. Other significant obstacles include the insufficient standardization of formulations, the complexity of regulatory frameworks for herbal medicines, and the need for comprehensive toxicity and pharmacokinetic studies [113]. Therefore, advancing toward the development of *T. officinale*-based therapies requires the implementation of rigorous protocols to ensure product quality, safety, and efficacy.

Future perspectives should focus on the design of well-controlled clinical trials to validate the therapeutic effects observed in preclinical models. Additionally, efforts should be directed toward the isolation and characterization of specific bioactive compounds with pharmacological potential. Furthermore, it is essential to deepen our understanding of the molecular mechanisms of action and optimize formulations to enhance their bioavailability and therapeutic effectiveness.

In conclusion, integrating these strategies may facilitate the translation of scientific knowledge into concrete clinical applications, offering new therapeutic alternatives based on *T. officinale.*

## 3. Literature Search Strategy and Selection Criteria

This narrative review was conducted to compile and critically analyze the scientific evidence regarding the biological and hepatoprotective properties of *Taraxacum officinale.* A literature search was carried out using three major scientific databases: PubMed, Scopus, and Web of Science, covering the period from 1973 to April 2024.

The search strategy included combinations of the following keywords and Boolean operators: 

“*Taraxacum officinale*” OR “dandelion” AND “liver” OR “hepatoprotection” OR “hepatotoxicity” OR “liver fibrosis” OR “steatosis” OR “antioxidant” OR “anti-inflammatory”. Inclusion criteria: peer-reviewed original articles, reviews, and ethnopharmacological reports; studies conducted in vitro, in vivo, and clinical trials; articles written in English or Spanish; studies focused on the effects of T. officinale or its isolated bioactive compounds on liver-related diseases or hepatocellular protection. Exclusion criteria: studies with insufficient methodological detail; articles not related to liver or hepatic function; non-scientific reports, news articles, or commercial reviews.

From the bibliographical review, approximately 200 articles were initially identified across selected databases. After applying inclusion and exclusion criteria and evaluating the relevance and quality of each study, around 100 articles were selected and included in this review. Data were extracted manually and organized into thematic sections addressing the plant’s phytochemistry, pharmacological mechanisms, and evidence for hepatoprotective activity.

## 4. Conclusions

*T. officinale* has several bioactive compounds that have caused considerable interest in the scientific community. Compounds that, when studied in vitro or preclinical studies, have been revealed to have anti-inflammatory, antioxidant, diuretic, and hepatoprotective properties. As a result, it has started to be evaluated as a treatment for diseases, mainly obesity, diabetes, osteoporosis, and cancer. In recent years, since the diagnosis of liver disease has increased dramatically, especially fatty liver and metabolic diseases, the research focused on the potential use of *T. officinale* in liver diseases has increased considerably. Most of this research has been developed in animal models or cell lines. In this way, scientists have been able to identify a variety of mechanisms of action, mainly antioxidant and anti-inflammatory responses. However, further research is still required to fully understand how *T. officinale* or its components interact molecularly with hepatic metabolic pathways and how these can influence metabolic homeostasis. To date, research has identified *T. officinale* as a potential therapeutic; however, detailed research is needed to establish its efficacy and safety in clinical trials. Optimal doses could be determined and interactions evaluated with other medications commonly used in patients with liver disease to improve their therapeutic effect.

## Figures and Tables

**Figure 1 pharmaceuticals-18-00990-f001:**
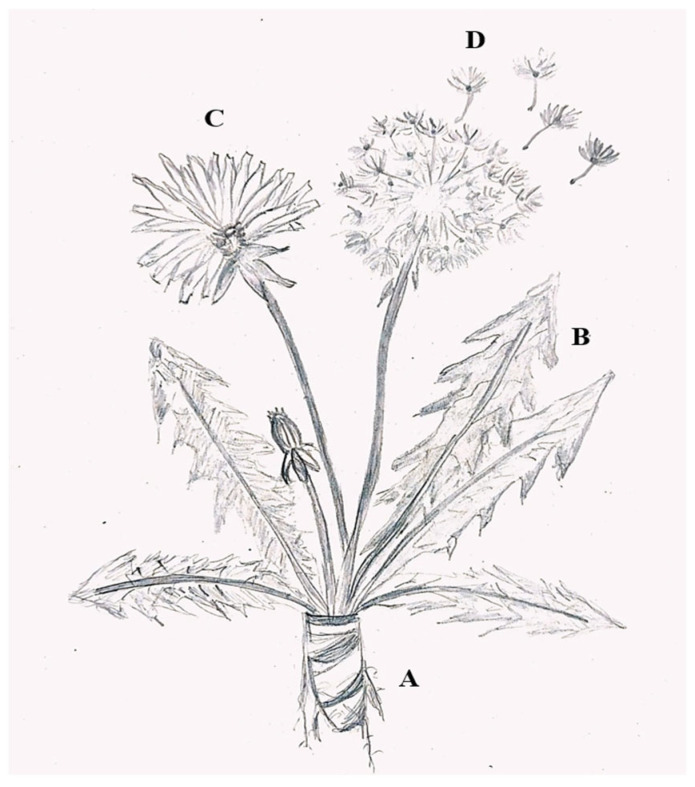
The morphology of *Taraxacum officinale*. (**A**) Roots; (**B**) leaves; (**C**) flower; (**D**) seed with fruit. (achene).

**Table 1 pharmaceuticals-18-00990-t001:** Main organic compounds identified in *T. officinale*, their anatomical origin, and associated biological activities reported in preclinical models.

Organic Compounds	Chemical Structure (Example)	Part of the Plant	Biological Activity	Example of Protective Effects	Refs.
Inulin(carbohydrate)	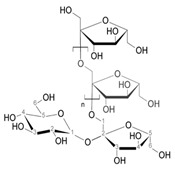	Root	Improves intestinal microbiota, anti-inflammatory, stimulates the elimination of pathogens.	An important carbohydrate such as inulin eliminates pathogens in the gastrointestinal tract and stimulates the suppression of obesity, cancer, and osteoporosis. In addition, presents diuretic activity increases choleretic function, anti-inflammatory action, and improves the intestinal microbiota.	[23,24]
Taraxacina (sesquiterpene lactones)	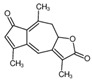	Root and aerial part	Anti-inflammatory, antimicrobial.	Anti-inflammatory and antimicrobial properties with greater effectiveness against Gram-positive strains. Appetite stimulator, hepato-renal drainer, and anti-inflammatory potential.	[10,24,25]
Luteolin chlorogenic acid and caffeic acid (Phenolic acids and flavonoids)	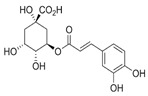	Root and aerial part	Anti-inflammatory, antioxidant, reduces lipid peroxidation and cellular damage.	The bioactive compounds present in *T. officinale*, such as luteolin, chlorogenic acid, caffeic acid, and terpenoids, exhibit notable anti-inflammatory, antioxidant, and hepatoprotective properties. These compounds contribute to liver and digestive system health by activating the Nrf2 transcription pathway in human hepatocytes, enhancing antioxidant defense mechanisms. They reduce oxidative stress by decreasing lipid peroxidation, protein carbonylation, and ROS production, particularly under ethanol-induced damage. Additionally, they downregulate the expression of inflammatory cytokines such as TNF-α and IL-6 and inhibit the nuclear translocation of NF-κB, a key regulator of inflammation. Some studies also suggest antihyperglycemic effects and potential antiviral activity against HBV through modulation of PTBP1 and SIRT1 expression levels	[6,14,26,27,28,29]
Taraxasterol(pentacyclic triterpene)	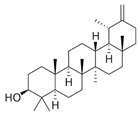	Root	Antioxidant, anti-inflammatory, antitumor.	Antihyperglycemic and anti-inflammatory properties. Decreased expression level of PTBP1 and SIRT1 proteins and may inhibit HBV and be a possible anti-HBV drug. In addition, it has antioxidant effects by reducing the production of ROS against oxidative damage caused by ethanol after improving the antioxidant enzymes present in the liver, in turn reporting an anti-inflammatory effect by reducing the production of TNF-α and IL-6 molecules.	[30,31]

Note: biological activity may vary depending on compound concentration, extraction method, and the physiological context in which it is evaluated. Abbreviations: Nrf2: nuclear factor erythroid-related factor 2; NF-kB: nuclear factor kappa-light-chain-enhancer of activated B cells; PTBP1: polyprime tract-binding protein 1; SIRT1: sirtuin-1 NAD-dependent deacetylase; ROS: reactive oxygen species; TNF-α: tumor necrosis factor alpha; TLR2: toll-like receptor 2.

**Table 2 pharmaceuticals-18-00990-t002:** Biological effects of *T. officinale* in various disease models.

Etiology	Associated Pathology/ Syndrome	Effect	Origin of the Extract	In Vivo*/*In Vitro Model	Refs.
	T1DM	Anti-diabetic/antioxidant	Leaf extract	Sprague-Dawley Rats (Males)	
	T2DM	Anti-diabetic/hypoglycemic	Full extract	T2DM patients; Albino Rabbits (males); INS-1 Cell Line; Sprague-Dawley Rats (Male)	[32,33,34,35]
			Leaf extract	IR-HepG2 Cell Line; inhibitory activity of α-amylase and α-glucosidase	
Metabolic disease			Leaf and root extract	Albino Rabbits (Male)	
	Obesity	Anti-obesity	Leaf extract	BALB/c Mice (Male)	[36]
		antihypertensive/antioxidant	Leaf and root extract	Wistar Rats and Swiss Mice (males and females)	[2]
		hypolipidemic/antioxidant	Leaf and root extract	Albino Rabbits (Males)	[37]
	Osteoporosis	anti-apoptotic/antioxidant/anti-inflammatory	Full extract Leaf extract	C57BL/6 Mice (Male) Albino Rats (Females)	[37]
					[38,39]
Pathogenic microorganism	Bacterial infection	Antimicrobial	Full extract	*S. aureus*, *E. coli*, and *L. monocytogenes*; *E. coli*, *S. aureus*, *K. pneumoniae*, and *P. mirabilis*	[25]
	Oxidative induction	Antioxidant	Full extract	V79-4 Cell Line and Chinese hamster (Males); *Oncorhynchus mykiss; MAC-T* Cell Line; Wistar Rats (Males)	[40,41,42,43]
			Extract leaves and petals	Human Blood	[29]
Oxidative stress	Cancerous cells	Antioxidant/hemostatic	Root extract	Human Blood	[44]
		antioxidant/antiproliferative	Full extract	HepG2 Cell Line	[45]
			Essential oil	Swiss Albino Mice (Males) and HeLa Cell Line	[46]
Inflammation	Induction of inflammation	Anti-inflammatory	Extract of flowers, roots, leaves, and bracts	* Chemical characterization of the compounds	[47]
			Leaf extract	RAW 264 Cell Line; C57BL/6 Mice (Male); Sprague-Dawley Rats (Male); Guinea Pigs (Females)	[48,49,50,51]
			Full extract	C57BL/6 Mice (Male)	[52]

Note: results may differ depending on the extract used, plant part, and experimental conditions. Clinical extrapolation should be approached with caution. * Chemical characterization of compounds without analysis of biological activity. Abbreviations: Diabetes Mellitus Type 1 (T1DM); Diabetes Mellitus Type 2 (T2DM); outbred mice derived from Swiss albino mice (ICR Mice); insulinoma-1 cells of rat (INS-1 Cells); insulin-resistant HepG2 cells of human (IR-HepG2 Cells); subline of the V79 Chinese hamster lung fibroblast (V79-4 Cell Line); primary bovine mammary alveolar cells (MAC-T); liver biopsy of a 15-year-old Caucasian male (HepG2 Cell Line); cervical cancer tumor of female human (HeLa Cell Line); tumor in a male mouse induced with the Abelson murine leukemia virus “A-MuLV” (raw264 Cell Line).

**Table 3 pharmaceuticals-18-00990-t003:** Hepatoprotective activity of *T. officinale* in liver disease.

Etiology	Associated Pathology/Syndrome	Effect	Origin of the Extract	In Vivo*/*In Vitro Model	Refs.
			Root extract	Wistar Rats (Males); Traditional Medicine (Humans); Rats and Mice; ICR Mice (Male); HepG2/2E1 Cell Line	[24,53,54,55,56,57,58,59]
			Taraxacol	ICR Mice; HepG cells 2.2.15. C57BL/6 Mice (Male). Chickens	[60,61,62,63]
	Induction of liver injury		Leaf extract	Wistar Rats (Males); Sprague-Dawley Rats (Male); Albino Mice (Male); C57BL/6 Mice (Male); C2C12 Cell Line	[40,64,65,66,67,68,69]
Liver disease			Root extract Root and leaf extract	Sprague-Dawley Rats (Male); C57BL/6 Mice (Male); Huh7 hepatoma Cell Line	[27,70]
		Hepatoprotection	Full extract	ICR Mice (Male); Kunming Rats (Male); Traditional Medicine (Humans); Sprague-Dawley Rats (Male)	[71,72]
			Polyphenol extract	Kunming Rats (Male)	[73]
			Infusion of leaves	Traditional Medicine (Humans)	[74,75,76]
	Hepatocellular carcinoma		Polysaccharides	Rats Sprague–Dawley (Male)	[77,78]
			Taraxacol	Human HCC Cell Lines (HepG2 and Huh7) and mouse HCC H22 Cell Line. Mouse BALB/c Kunming Mice	[62]
			Apigenin	Hep3B Cell Line	[79]

Note: efficacy may depend on extract type, dosage, and duration of treatment. Clinical studies are needed to confirm these effects in humans. Abbreviations: liver cancer human cell that express the cytochrome P450 2E1 “CYP2E1” (HepG2/2E1 Cell Line); isolated from a hepatoblastoma of human (HepG cells 2.2.15.); isolated from a hepatoma of male human (Huh7 Cell Line); isolated from mouse hepatocellular carcinoma (HCC cells H22); isolated from mouse hepatocellular carcinoma (HCC cells H22); isolated from a liver cancer of an 8-year-old black male (Hep3B Cell Line).

## Data Availability

No new data were created or analyzed in this review. Data sharing is not applicable to this article.

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
