# Peer review of "The Role of Dandelion (Taraxacum officinale) in Liver Health and Hepatoprotective Properties"

_pharmaceuticals, 2025, doi:10.3390/ph18070990_

Round 1

Reviewer 1 Report

Comments and Suggestions for Authors

This document reviews the scientific evidence on the effects of dandelion, a plant widely used in traditional medicine to treat liver disorders. The article compiles and critically analyzes the existing literature on the biological and hepatoprotective properties of Taraxacum officinale. To do this, the authors conducted a systematic search of high-impact databases, including PubMed, Scopus, and Web of Science, covering publications from 1973 to April 2024. The review covers the plant's phytochemical composition, historical uses, and the in vitro and in vivo experimental evidence of its biological activity. A key finding is the consistent preclinical evidence showing that T. officinale extracts and isolated bioactive compounds, such as taraxasterol, protect the liver from damage induced by toxicants (e.g., acetaminophen and alcohol), fibrosis, and oxidative stress.

The document is well-written. The "Results" section appropriately consists of a synthesis of existing literature. The information is logically organized into sections—botanical description, chemical composition, and biological evidence—and is supported by tables that effectively summarize key findings. The authors interpret the data cautiously, describing the preclinical findings as "promising" while avoiding exaggerated claims. They correctly acknowledge that most research has been conducted in animal models or cell lines and highlight the need for human validation, demonstrating a responsible interpretation of the available evidence.

Overall, the manuscript is presented in clear, academic language. Its logical structure and argumentation are easy for a knowledgeable audience to follow. Therefore, I recommend that the paper be accepted for publication, provided the following minor corrections are made:

  • On the first page, the author list ends with “Jessica Zúñiga and ernandez 1*.” The last name “ernandez” is not capitalized and appears to be incomplete or misspelled.
  • As a stylistic recommendation, the text in the Abbreviation tables should be left-aligned for better readability.
  • In the abbreviation table on page 12, the abbreviation for “Inhibitor alpha of kappa B” is listed as “Ικ-Βα.” This should be corrected to the standard abbreviation, “IκBα.”

Author Response

Dear Reviewer

Pharmaceuticals

MDPI

    My colleagues and I have taken into consideration the recommendations by the reviewers to our manuscript entitled “The Role of dandelion (Taraxacum officinale) in Liver Health and Hepatoprotective Properties” (pharmaceuticals-3709654). We would like to sincerely thanks the reviewers for their thorough and constructive comments, which have helped us to significantly improve the quality and clarity of our manuscript. We carefully considered each suggestion and have revised the manuscript accordingly. Below, we provide a detailed point-by- point response outlining the changes made in response to each comment. All modifications are clearly indicated in the revised version of the manuscript.

Comments 1: On the first page, the author list ends with “Jessica Zúñiga and ernandez 1*.” The last name “ernandez” is not capitalized and appears to be incomplete or misspelled.

Reply 1: The author name was corrected to “Zúñiga-Hernández”. Line 5: Changed to “Jessica Zúñiga-Hernández”.

Comments 2: As a stylistic recommendation, the text in the Abbreviation tables should be left-aligned for better readability.

Reply 2: The alignment was corrected for better readability: Line 464 onward: Left alignment applied to all abbreviation entries.

Comments 3: In the abbreviation Table on page 12, the abbreviation for “Inhibitor alpha of kappa B” is listed as “Ικ-Βα.” This should be corrected to the standard abbreviation, “IκBα.”

Reply 3: Thanks for the comment, the abbreviation was corrected according to the standard format. Corrected to “IκBα” in abbreviation

Best regards

Jessica Zúñiga-Hernández, PhD.

Laboratorio de Farmacología

Departamento de Ciencias Básicas Biomédicas

Facultad de Ciencias de la Salud

Universidad de Talca,

Talca, Chile
email: jezuniga@utalca.cl

Reviewer 2 Report

Comments and Suggestions for Authors

The review article by Francisca Herrera Vielma et al. entitled ‘The Role of dandelion (Taraxacum officinale) in Liver Health and Hepatoprotective Properties’ focuses on the hepatoprotecive roles of dandelion extract. The review is extensive and elaborate with clear figure and tables. However, the following minor changes can be included to further improve the article.  

  1. The article highlights the plant extract overall. Including chemical structures of representative compounds like Inulin, Taraxasterol etc. would give more impact to the review.
  2. In table 1 one of the label officinale structures can be changed to plant parts for clarity.
  3. Even though the beneficial effects of the extracts/plant components/isolates are known for more than 2 decades as per the references cited, the reason for lack of any approved drug/therapy can be included in the discussion.
  4. The future prospects seems to be missing in the review which can be of interest to the authors.
  5. Any detrimental/side effects of the plant extracts (if any) can be included in the review.

Author Response

My colleagues and I have taken into consideration the recommendations by the reviewers to our manuscript entitled “The Role of dandelion (Taraxacum officinale) in Liver Health and Hepatoprotective Properties” (pharmaceuticals-3709654). We would like to sincerely thanks the reviewers for their thorough and constructive comments, which have helped us to significantly improve the quality and clarity of our manuscript. We carefully considered each suggestion and have revised the manuscript accordingly. Below, we provide a detailed point-by- point response outlining the changes made in response to each comment. All modifications are clearly indicated in the revised version of the manuscript.

Comments 1: The article highlights the plant extract overall. Including chemical structures of representative compounds like Inulin, Taraxasterol etc. would give more impact to the review.

Reply 1: We have accordingly done, and a Representative structures (e.g., inulin, taraxasterol) were added in Table 1

Comments 2: In table 1 one of the label officinale structures can be changed to plant parts for clarity.

Reply 2: The table was labeling accordingly. Line 172: The title of Table 1 was improving to “Main organic compounds identified in T. officinale, their anatomical origin, and associated biological activities reported in preclinical models” to improve clarity and readability. Additionally, the column previously titled “T. officinale structures” was changed to “Part of the plant” for greater precision, and a new column, “Chemical structure (example),” was added to illustrate representative molecular structures of the compounds.

Moreover, lines 180 and 189: The titles of Table 2 and Table 3 were revised to improve clarity and specificity. Table 2 was changed from “T. officinale in relevant pathologies” to “Biological effects of T. officinale in various disease models” (line 180), and Table 3 was changed from “T. officinale in hepatic pathologies” to “Hepatoprotective activity of T. officinale in liver disease” (line 189). These modifications aim to better reflect the content and scientific focus of each table.

Comments 3: Even though the beneficial effects of the extracts/plant components/isolates are known for more than 2 decades as per the references cited, the reason for lack of any approved drug/therapy can be included in the discussion. And The future prospects seems to be missing in the review which can be of interest to the authors.

Reply 3: thanks for your recommendations, a paragraph was added discussing this translational gap. Also, a paragraph discussing future research directions was added. Line 389-416: Added in “2.11 Limitations and future perspectives”.

Comments 4: Any detrimental/side effects of the plant extracts (if any) can be included in the review.

Reply 4: Safety and contraindications were included in a dedicated section. Lines 368–390: Addressed in “2.10”

Best regards

Jessica Zúñiga-Hernández, PhD.

Laboratorio de Farmacología

Departamento de Ciencias Básicas Biomédicas

Facultad de Ciencias de la Salud

Universidad de Talca,

Talca, Chile
email: jezuniga@utalca.cl

Reviewer 3 Report

Comments and Suggestions for Authors

The common dandelion (Taraxacum officinale) is a flowering plant (angiosperm) belonging to the Asteraceae family. It is widely used for medicinal purposes, as it contains a large number of pharmacologically active compounds, such as flavonoids, terpenoids, triterpenes, sesquiterpenes. Dandelion is attributed with anti-inflammatory, hypoglycemic, pancreatic stimulant and cholesterol-lowering properties. Furthermore, its extracts are used as liver purifiers, decongestants and detoxifiers. The present review focused on the biological activity of dandelion, with particular attention to its hepatoprotective effects. Therefore, a collection and analysis of published studies on the effects of T. officinale in animal models and on its potential application in liver diseases were performed.

The medicinal properties of dandelion have been known since ancient times. Therefore, the topic discussed is very current, as in recent times there has been a growing tendency to re-evaluate and deepen the health effects of herbal drugs and their preparations, also due to the increasingly frequent appearance of chronic diseases.

Overall, the structure of the manuscript is well organized and the various sections follow a logical sequence. However, some changes are required, as reported below.

Keywords: Please review the keywords and replace those already present in the title or repetitive with other suitable ones, such as: choleretic, cholagogue, antioxidant, anti-inflammatory.

At the end of the Introduction section, authors should report the purpose of their investigation with respect to existing publications.

Line 62. Rename the paragraph title, the current one is more suitable for an experimental article than a review.

Please replace Figure 1 with a clearer one.

The Tables are intended to make it quicker to understand the data without the aid of the text. Modify the captions to make them clearer and more specific. Footnotes should refer to each individual table.

Line 338. Rename the paragraph title, the current one is more suitable for an experimental article than a review.

For completeness of exposition, in addition to the beneficial effects, the authors should add the contraindications and interference with drugs resulting from the intake of dandelion.

Author Response

Dear Reviewer

Pharmaceuticals

MDPI

    My colleagues and I have taken into consideration the recommendations by the reviewers to our manuscript entitled “The Role of dandelion (Taraxacum officinale) in Liver Health and Hepatoprotective Properties” (pharmaceuticals-3709654). We would like to sincerely thanks the reviewers for their thorough and constructive comments, which have helped us to significantly improve the quality and clarity of our manuscript. We carefully considered each suggestion and have revised the manuscript accordingly. Below, we provide a detailed point-by- point response outlining the changes made in response to each comment. All modifications are clearly indicated in the revised version of the manuscript.

Comments 1: Keywords: Please review the keywords and replace those already present in the title or repetitive with other suitable ones, such as: choleretic, cholagogue, antioxidant, anti-inflammatory.

Reply 1: Thank you for pointing this: keywords were updated to avoid redundancy and include suggested terms. Line 35-37: Final keywords: bioactive compounds; traditional medicine; taraxasterol; antioxidant; liver fibrosis; chronic liver diseases.

Comments 2: At the end of the Introduction section, authors should report the purpose of their investigation with respect to existing publications.

Reply 2: Thanks, the aim statement was added. Line 128-133: Added sentence beginning with “Therefore, the aim of this narrative review is to critically synthesize the current preclinical evidence regarding the hepatoprotective effects of T. officinale, focusing on its bioactive compounds mechanisms of action, and potential applications in liver disease. Unlike previous reviews that broadly cover the general medicinal uses of dandelion, this work specifically highlights its role in liver protection and metabolic liver disorders, emphasizing recent findings published in the last two decades.”

Comments 3: Line 62. Rename the paragraph title, the current one is more suitable for an experimental article than a review.

Reply 3: As you proposed, the title was rephrased for appropriateness in a review article. Line 63: The section title was changed from “Results” to “2. Botanical, Ethnopharmacological and Experimental Overview of Taraxacum officinale” to better reflect the structure and content of the review.

Comments 4: Please replace Figure 1 with a clearer one.

Reply 4: A higher-quality figure was inserted. Line 81: The figure was replaced.

Comments 5: The Tables are intended to make it quicker to understand the data without the aid of the text. Modify the captions to make them clearer and more specific. Footnotes should refer to each individual table.

Reply 5: Captions and footnotes were revised for clarity. The footnotes of Table 1 (lines 175–179), Table 2 (lines 181–188), and Table 3 (lines 190–196) were revised to include not only abbreviations, but also an explanatory footnote to improve clarity.

Comments 6: Line 338. Rename the paragraph title, the current one is more suitable for an experimental article than a review.

Reply 6: We agree with this comments, the title was revised to better reflect content of a review. Line 415: The section title was changed from “Materials and Methods” to “Literature Search Strategy and Selection Criteria” to more accurately reflect the content and purpose of the section.

Comments 7: For completeness of exposition, in addition to the beneficial effects, the authors should add the contraindications and interference with drugs resulting from the intake of dandelion.

Reply 7: A dedicated section discussing safety and interactions was added. Lines 365-387: New section “2.10 Safety Considerations, Contraindications, and Interactions of Taraxacum officinale”

Best regards

Jessica Zúñiga-Hernández, PhD.

Laboratorio de Farmacología

Departamento de Ciencias Básicas Biomédicas

Facultad de Ciencias de la Salud

Universidad de Talca,

Talca, Chile
email: jezuniga@utalca.cl